PERSPECTIVE

# Prebiotic chemical origin of biomolecular complementarity

Y. Sajeev [1,2] ✉

The early Earth, devoid of the protective stratospheric ozone layer, must have sustained an ambient prebiotic physicochemical medium intensified by the co-existence of shortwave UV photons and very low energy electrons (vLEEs). Consequently, only intrinsically stable molecules against these two co-existing molecular destructors must have proliferated and thereby chemically evolved into the advanced molecules of life. Based on this view, we examined the stability inherent in nucleobases and their complementary pairs as resistance to the molecular damaging effects of shortwave UV photons and vLEEs. This leads to the conclusion that nucleobases could only proliferated as their complementary pairs under the unfavorable prebiotic conditions on early Earth. The complementary base pairing not only enhances but consolidates the intrinsic stability of nucleobases against short-range UV photons, vLEEs, and possibly many as-yet-unknown deleterious agents co-existed in the prebiotic conditions of the early Earth. In short, complementary base pairing is a manifestation of chemical evolution in the unfavorable prebiotic medium created by the absence of the stratospheric ozone layer.

The biomolecular complementarity that begins with the chemical evolution of DNA base pairs as fundamental components of advanced molecules to replicate and transfer genetic information is one of the most crucial developments in the emergence of life on Earth. These hydrogen bonded complementarity have long been believed to provide significant thermodynamic stability to the double helix structure of DNA. In contrast to this perception, modern experiments have shown that the thermodynamic stability of the duplex is due to $\pi$-stacking, rather than to the hydrogen bonded complementary base pairing[1]. As the hydrogen bonded complimentary base pairing does not contribute to the thermal stability of the duplex structure, their consistent structural composition throughout the evolution raises an intriguing question: How did complementary base pairs evolve chemically as the basic unit of life on early Earth?

If the hydrogen bonding between the base pairs does not contribute to the thermodynamic stability of their advanced molecules, then a robust explanation of the molecular origin of biocomplementarity may necessitate a clear understanding of the physico-chemical conditions of the prebiotic environment within which their chemical evolution occurred. Fossil evidence suggests that the earliest forms of life appeared on Earth as early as 3.7 billion years ago (bya)[2-4]. Although the primitive life on Earth have originated 3.7 bya, the atmospheric oxygen, which is known to arise from the Earth's biophotosynthesis by cycanobacterium, did not have formed until very late about 2.4 bya[5]. Consequently, chemical evolution of life-forming advanced molecules may have taken place long before the formation of ozone in the stratosphere, i.e., abiogenesis occurred under intense shortwave UV conditions[6]. Therefore, to comprehend the evolution of complementary base pairs as the fundamental units of life, it is necessary to discuss prebiotic molecular processes that may have taken place under intense shortwave UV photons. The perspective that emerges from such a discussion is presented below.

[1] Theoretical Chemistry Section, Bhabha Atomic Research Centre, Mumbai, India. [2] Homi Bhabha National Institute, Mumbai, India. ✉email: sajeevy@barc.gov.in

## Main

**Synergism of molecular mass-growth inhibitors.** Shortwave UV (<300 nm) is a highly active part of solar radiation and it is supposed to have reached the prebiotic medium unattenuated[7]. Such intense shortwave UV conditions directly impede the proliferation of prebiotic biomolecular precursors by causing them structural damages. Therefore, the photophysical resistance to structural damage from intense shortwave UV radiation must have been the most decisive factor in the proliferation of prebiotic building units and their chemically evolved molecular structures[7]. Consequently, only the fundamental molecules that are intrinsically stable against the structural damages caused by shortwave UV photons are proliferated, and progressed through chemical evolution into advanced molecules of life. While recognizing the paramount importance of this intrinsic photostability of prebiotic molecules in chemical evolution, one may ask a fundamental question: Is the proliferation of chemically evolved prebiotic molecules subject only to their photochemical stability?

Very low-energy electrons (vLEEs), i.e., free-electrons at energies below subexcitation energy (<3 eV), are another reactive species as important and ubiquitous as shortwave UV photons under prebiotic conditions. The importance of vLEEs in the context of chemical evolution comes from the fact that they are one of the ubiquitous chemically active species whose resonant capture into the molecular electron field can trigger molecular demolition[8,9]. Although there are many phenomena that allow the production of vLEEs, secondary ionization of molecules occurring in the tracks of high energy ionizing radiation such as X-rays and gamma-rays and the tracks of electric discharge are believed to be the main reason for their production at high quantities under prebiotic conditions. However, since these high-energy radiation tracks do not provide a favorable environment for the proliferation of enough prebiotic molecules, the direct role of secondary electrons in the chemical evolution of prebiotic molecules has remained largely unexplored. The role of vLEEs in prebiotic chemical evolution has become relevant following a recent report that large numbers of vLEEs are efficiently produced even upon the absorption of ambient-intensity UV photons in a system abundant with $\pi$-molecules[10].

When multiple subunits of a molecular system absorb UV radiation, the system relaxes via ejecting free electrons through a mechanism known as intermolecular Coulombic decay (ICD)[11–16]. This non-local ionization process is very efficient when the intermolecular interactions are activate between excited molecules. Most interestingly, in electronically excited $\pi$-molecular systems, because of the intermolecular covalent interaction between the diffused $\pi^*$ orbitals, a very efficient and fast ICD occurs, and as a consequence vLEEs are generated, even at ambient UV intensities. The fact that copious amounts of vLEEs are naturally produced by UV irradiation in a prebiotic medium rich in $\pi$-molecular systems, including chemically evolved molecules such as nucleobases, suggests their critical role in determining the course of prebiotic chemical evolution. In short, life on Earth has evolved under the constant presence of many deleterious agents, such as intense shortwave UV photons and ubiquitous vLEEs.

**Synthesis vs proliferation of prebiotic molecules.** Molecular media receiving a constant dose of intense shortwave UV photons, similar to the prebiotic conditions of the Earth, are consequently transformed into a hotbed of vLEEs and many other secondary reactive species. The classic Miller-Urey experiment and its many subsequent variants have shown that such a prebiotic open medium, continuously receiving energy, can synthesize all kinds of precursor and building block molecules through its simple reactants[17–19]. Although such an open prebiotic medium produces molecular building blocks from its simple reactants, such synthesized molecules are more susceptible to molecular damages by the deleterious agents of the same medium than their simple precursor reactants[20]. Since fossil studies indicate that advanced molecules of life arose from the accumulation of such synthesized organic product molecules over a long prebiotic period[21], the proliferation of these product molecules under highly adverse prebiotic conditions created by the coexistence of various deleterious agents must have been the greatest challenge in chemical evolution. Consequently, it can be correctly assumed that the proliferated molecules were intrinsically stable against the coexisting shortwave UV photons and vLEEs, and these molecules were chemically advanced by their increased prevalence over time.

**Proliferation and unified intrinsic stability (UIS) mechanisms.** Starting with the classic Miller-Urey's experiment, there have been several attempts to recreate prebiotic chemical evolution[17–19]. A recent Miller-Urey model experiment even succeeded in synthesizing nucleobases under an extreme physicochemical condition set by the high-power shock wave plasma[19]. Apparently, this proliferation of nucleobases underscores a remarkable integrated stability mechanism inherent in them against the harmful molecular deleterious agents that existed in the harsh medium. Furthermore, in the context of our assumption that the prebiotic medium was intensified by the coexistence of shortwave UV photons and vLEEs, the emergence of nucleobases under extreme physicochemical conditions of the Miller-Urey model experiment raises a very intriguing question: Is such an integrated stability mechanism against shortwave photons and vLEEs inherent in the nucleobases or in their chemically advanced molecules?

Our discussion is based on the assumption that the prebiotic medium was intensified by the shortwave UV photons and vLEEs. In such a prebiotic medium, chemically evolved nucleobases and their advanced molecules are susceptible to structural loss mainly due to UV absorption or resonance attachment of vLEE. Proliferation of prebiotic molecules, as opposed to this impending structural loss, implies that the molecules were internally stabilized by rapid molecular processes that were faster than nuclear dynamics causing molecular damage. As will be discussed shortly, such rapid molecular processes are generally originate from a common chemical characteristic of the advanced molecules–the population of their $\pi^*$-orbital. If the mechanisms preventing such structural loss are inherent in these molecules, they must also originate from the topology of the corresponding electronic states populating the $\pi^*$ orbital. This implied that the preventive mechanisms against the structural damages caused by shortwave UV photons and vLEEs should be similar. We refer to these similar intrinsic mechanisms that stabilizes molecules rapidly against multiple major molecular deleterious agents as unified intrinsic stability (UIS). Most importantly, continual process of refinement of UIS manifested at each step in the progression of the chemical evolution may have determined the structural composition of the advanced molecules proliferated on early Earth. Therefore, UIS mechanisms that exist in building block molecules and their refinement through their advanced molecular structures are direct indicators to a prebiotic regime. A comprehensive study of the UIS of the functional moieties of evolved molecules against potential deleterious agents under various possible prebiotic conditions may reveal unknown stages of their chemical evolution on the early Earth.

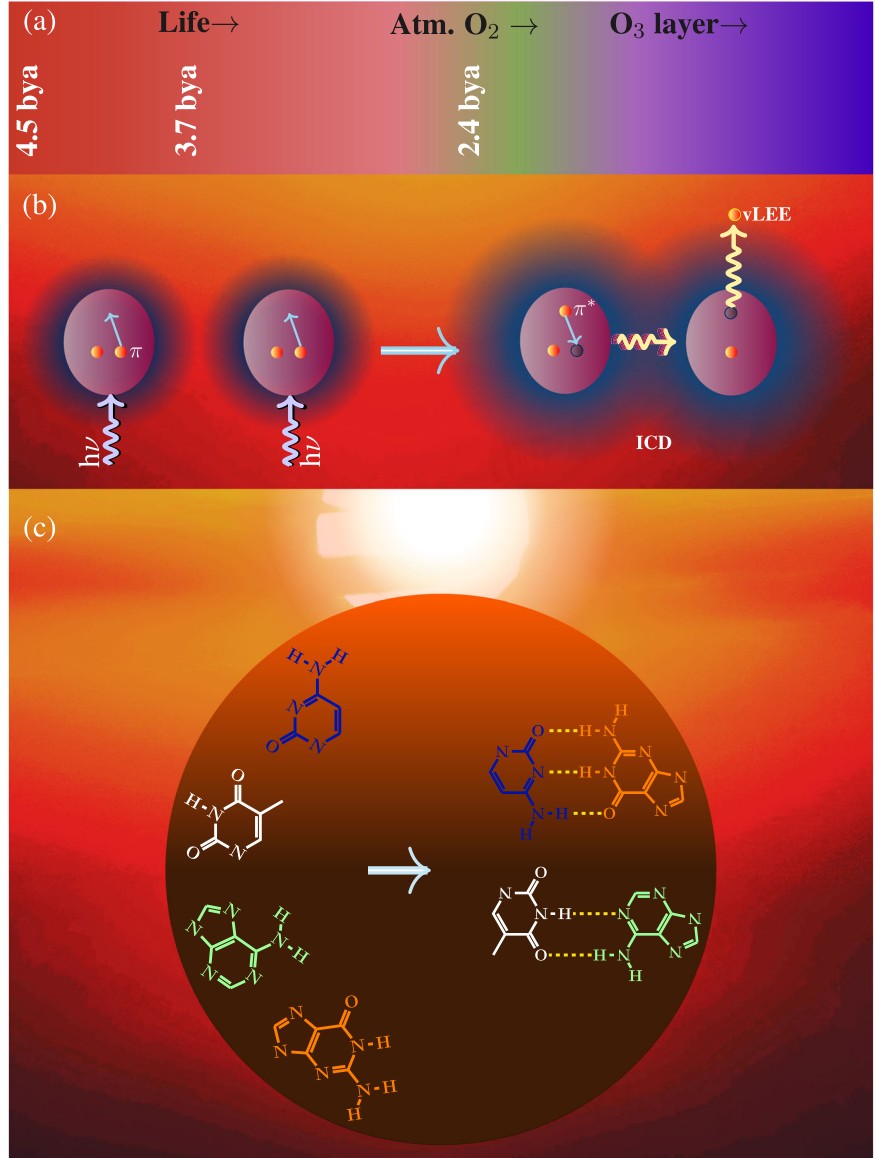

**Fig. 1 Chemical evolution of complementary base pairs in a UV and vLEE intense prebiotic physicochemical condition.** The color palate of (**a**) illustrates the evolution of the stratospheric ozone layer. The intensity of the purple color indicates the photon density of shortwave surface UV radiation as well as the abundance of vLEEs produced by UV radiation. While the green color represents atmospheric oxygen, the blue color indicates the appearance of stratospheric ozone layer. The panel (**b**) shows the ICD mechanism which produces vLEE from two nearby $\pi\pi^*$ photoexcited molecules. Because of the non-local interactions, $\pi^*\pi$ electronic deexcitation in one of the photoexcited molecule attributes to the ionization of its photoexcited neighbor. The proliferation of complementary base pairs under UV- and vLEE-rich prebiotic conditions on early Earth is shown in (**c**).

**Search for UIS mechanisms in biomolecules**. Among the various deleterious agents that coexisted under the physicochemical conditions of the prebiotic medium, shortwave UV photons and very low-energy electrons unleash the most efficient molecular damage by directly perturbing the outer-valence electrons of nucleobases. Therefore, UIS mechanism against the molecular damages by shortwave UV photons and very low-energy electrons must have become the most important in determining the proliferation of advanced molecules of life. In spite of billions of years of chemical evolution, life must have retained some direct remnants of the molecular mechanisms underlying the UIS mechanism against shortwave UV photons and vLEEs. Since the complementary base pairs are considered to have remained unchanged throughout evolution of life on Earth[22], the molecular imprints of the prebiotic chemical evolution should first sought within these molecules. Another important reason for choosing

complementary base pairs to identify the existence of UIS is that the fast non-enzymatic molecular processes specific to complementary base pairs are known to deactivate exogenously induced reactive states of DNA that require a rapid response[23]. If the complementary pairs are intrinsically stabilized against shortwave UV photons and vLEEs through similar molecular mechanisms, the existence of such a UIS would be the first fundamental molecular indication to the manifestation of a disruptive prebiotic medium in chemical evolution. Therefore, we follow a reductionist examination of the inherent abilities of complementary pairs against UV photons and vLEEs. An illustration of this theme is presented in Fig. 1.

**Nucleobases are intrinsically photostable!**. As a prelude to a discussion on the existence of unified intrinsic stability (UIS) in

complementary pairs, we first review the known intrinsic photochemical stability of individual nucleic acid bases. Nucleobases are the most fundamental biomolecular chromophores that absorb shortwave UV radiation. Absorption of UV radiation below 320 nm wavelength by a DNA nucleobase, which populates its $\pi\pi^*$ excited state, can trigger a variety of photochemical reactions in the DNA leading to mutagenesis and carcinogenesis[24]. However, such fatal consequences of UV absorption by the DNA chromophores are largely avoided due to an intrinsic photostability of DNA bases[25,26]. An ultrafast radiationless decay in the photoexcited nucleobase moiety retains its electronic ground state before any photochemical reaction takes place[25]. This is a result of the conical intersections connecting the $\pi\pi^*$ excited state to the electronic ground state through an intermediate $\pi^*\sigma^*$ state[26]. This facile internal conversion that makes the excited state lifetime ultra-short (A: 1.0 ps; G: 0.8 ps; T: 6.4 ps: C: 3.2 ps)[25], and provides DNA bases a high degree of intrinsic photostability. In fact, this inherent molecular stability of the nucleobase must have been indispensable for their proliferation in the prebiotic chemical environment. Even more intriguing is the chemical evolution of their complementary base pairs as the fundamental units as durable carriers of genetic information.

**Complementary pairing enhances intrinsic photochemical stability!.** When UV photons are absorbed by a DNA base, the corresponding excited state must return to a ground state quickly to avoid harmful reactions. The production of mutagenic photoproducts due to UV absorption are known to be avoided through a rapid excited-state-deactivation path involving the interpair transfer of a proton in one of the interstrand hydrogen bonds[27–29]. During this photodeactivation mechanism, which is specific to the complementary base pairing, the energy from the UV irradiation is transferred into nuclear modes which allow the molecule to return to the ground state.

A schematic representation using the potential energy profile of the molecular mechanism behind this excited state deactivation in the model compound 2-aminopyridine (2AP) is depicted in Fig. 2. While the $^1\pi\pi^*$ excited-state population of the isolated monomer of 2AP relaxed with a lifetime ($\tau$) of 0 1.5 ns, the corresponding locally excited $^1\pi\pi^*$ of its doubly hydrogen-bonded complementary dimer relaxed with a remarkably reduced lifetime of 65 ps, proving the existence of an additional relaxation pathway specific to the complementary pair[29]. Thus, the opening of new ultrafast deactivation pathways due to the base pairing bestows enhanced photostability on base pairs and, hence, preserve the structural integrity of the genetic code following absorption of shortwave UV radiation. Most remarkably, among all the tautomeric forms of base pairs, this intrinsic photostability is a characteristic for the complementary base pairs[27,28].

The protective function of the $^1\pi\pi^*$ specific to the base pair becomes more relevant in a protic medium. Although isolated nucleic acid bases are intrinsically photostable, the presence of an aqueous environment may prevent the ground state recovery of its $\pi^*$ excited state, where the $^1\pi\sigma^*$ states promote a hydrogen-transfer process from the photoexcited nucleobase to the protic solvent[26]. Subsequent burial of the transferred hydrogen in the solvated medium can lead to molecular damage. However, when the chromophore base is complemented in a base pair through intermolecular hydrogen bonds, the ground state recovery via a newly created $^1\pi^*\pi^*$ path along the interpair hydrogen transfer can compete with the hydrogen atom's burial into the medium. This may be another reason why base pairs evolved as the basic units of the genetic code.

**Canonical vs. non-canonical complementary pairs.** One of the most important puzzles in the origin of life is selection in chemical evolution confined to purine-pyrimidine canonical base pairs[30]. The fact that non-canonical complementary base pairs such as purine-purine and pyrimidine–pyrimidine base pairs have not evolved into advanced molecules of life is also an apparent refutation of the conventional notion that the thermodynamic stability provided by hydrogen bonding was the key to chemical evolution. Although purine-pyrimidine complementarity is justified by the geometry of the double-stranded helix, in light of the reports that duplex structures suitable for information storage and replication fundamental to molecular evolution are possible from purine bases alone[31], the question of why Nature does not select such complementary structures becomes relevant. It is worth asking if the UIS mechanisms are responsible for prebiotic selection of canonical pairs.

One can attribute the rapid recovery of the ground state of photoexcited complementary pairs to a potential energy curve crossing between the ultrafast intermolecular-proton transfer path of the corresponding state and the neutral ground state (see Fig. 2). Complementary pairing between non-identical molecules, such as canonical base pairing, enables such curve crossing easily without much geometrical distortion of the molecules other than the proton transfer. On the other hand, in non-canonical symmetric pairs, i.e., pyridine-pyridine pairs and purine-purine pairs, the corresponding proton transfer path is symmetric double well and does not cross with the neutral ground state. A strong non-planar structural distortion of the moieties is essential for the proton transfer path to cross with the neutral ground state. Since these non-planar structural distortion can cause strand ruptures when the sugar-phosphate units bind to base pairs, the advanced molecules of non-canonical complementary pairs involving backbone units may not have proliferated under the harsh conditions of the prebiotic medium.

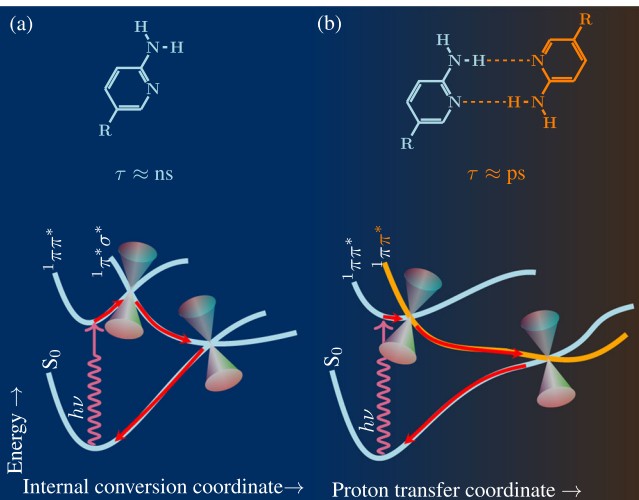

**Fig. 2 Enhanced excited state deactivation due to pairing.** In (**a**), the potential energy profiles identify the mechanism by which the excited state is deactivated in a monomer. The photo-excited $^1\pi\pi^*$ state intersects with a dissociative $^1\pi\sigma^*$ state. Through an ultrafast internal conversion due to a conical intersection that connects the $^1\pi\sigma^*$ with the ground state, the excited molecule regains its electronic ground state before any photodissociation damage occurs. In (**b**), enhancement in the deactivation of a locally excited $^1\pi\pi^*$ state due to the opening of an new ultrafast electron-driven proton transfer channel between the monomers is shown (see ref. [29]).

**Stability against vLEEs.** The principal role of photochemical stability in chemical evolution has long been known[7,20,29].

However, the role of multifaceted indirect effects of UV radiation on prebiotic chemical evolution has not been sufficiently discussed. The most important indirect effects of shortwave UV radiation in a chemical medium is the production of vLEEs[10,32–34]. Many photochemical channels have been recently reported that enable the efficient production of vLEEs as a result of shortwave UV irradiation in medium containing $\pi$ molecules[10,32,33]. In the reported ICD experiments of refs. [10] and [32], LEEs and their counterpart cations produced by $\pi$-molecules under ambient UV light via the ICD mechanism were confirmed by a set of experiments that rule out the role of photogenerated electrons, rather than a coincidence measurement of the electron-cation pair. The significance of these channels in the chemical evolution of advanced molecules lies in the fact that the prebiotic environment is also rich in $\pi$-molecular systems, including nucleic acids. That is, vLEEs must have been abundantly produced in the prebiotic medium by intense shortwave UV photons. Therefore, the chemical changes caused by the interaction of vLEEs with prebiotic molecules should be considered as one of the main indirect effects of UV radiation on the chemical evolution of advanced molecules.

Very low-energy electrons interact most efficiently with molecules via resonant attachment/capture. Many functional moieties of biological macromolecules are amenable to resonant capture by vLEEs. Among them, nucleobases are most susceptible to the resonant capture[35]. Their low-energy $\pi^*$ orbitals are very efficient resonance doorways for the capture of vLEEs[36–38]. This resonant capture of vLEEs into the low-energy $\pi^*$ orbitals of nucleobases has one of the highest electron capture cross-sections known[35]. When a vLEE is resonantly captured by a nucleobase moiety, a metastable compound electronic state of the vLEE and the nucleobase moiety, often referred to as the negative ion resonance state (NIRS) of the moiety, is formed. The $\pi^*$-NIRS of isolated nucleobases decay to the ground state primarily by autodetachment[35]. This means, just as nucleobases are intrinsically stable to survive $\pi^*$ photoexcitation, they are also structurally stable against resonance capture of vLEEs into $\pi^*$ orbitals. However, the low energy N-glycosidic $\sigma^*$ anti-bonding

orbital of the nucleotide can cross with the $\pi^*$ orbital for a short stretch of the corresponding bond. As a result, the resonantly captured $\pi^*$ electron can easily get transferred to the $\sigma^*$ orbital or competitively to the $\sigma^*$ orbital of the CX-O bond (where $X = 5$ or 3)[39]. The corresponding C-N $\sigma^*$ populated electronic state, i.e., $^2\Sigma_u$ NIRS, is a dissociative state. These dissociative NIRS are known to relax through two highly efficient bond-breaking processes (see Fig. 3a)[36–38]. The former is an ionic fragmentation channel known as dissociative electron attachment (DEA)[40], while the latter is a neutral dissociation channel known as bond breaking by catalytic electron (BBCE)[8,41]. The population of $^2\Sigma_u$ NIRS of the nucleobase moiety leads to the DEA fragmentation at the N-glycosidic bond[42] or at the sugar-phosphate backbone[43]. That is, vLEEs, in principle, can unleash the most devastating molecular damage that destroys the very structural integrity of advanced molecules of nucleobases.

In an adverse prebiotic medium, where resonant interactions with vLEEs are ubiquitous, molecular mechanisms that intrinsically protect the structural integrity of advanced molecules of nucleobases must be essential for their proliferation. Moreover, in such a medium rich in vLEEs and shortwave UV photons, an internal stability achieved by a same molecular mechanism against assaults caused by both these co-existing deleterious agents is also essential for the efficient proliferation of advanced molecules. As in the case of photostability, this situation necessitates the investigation of whether the base pairing also confers structural stability to nucleobases against highly efficient $\pi^*$-type resonant capture of vLEEs. Therefore, we discuss here the possibility of fast non-damaging nuclear relaxation channels of base pairs that facilitate the autoionization of resonantly captured vLEE from its molecular field.

The vLEE-biomolecular interaction has recently undergone extensive research to understand the chemical reactions that cause genetic damage. Whether these studies point to unique stability of base pairs against vLEEs is of great relevance here. Striking differences have been emerged between vLEE-induced damage to oligonucleotide and double-stranded DNA[44,45]. The cross-section of vLEE-induced single-strand break is

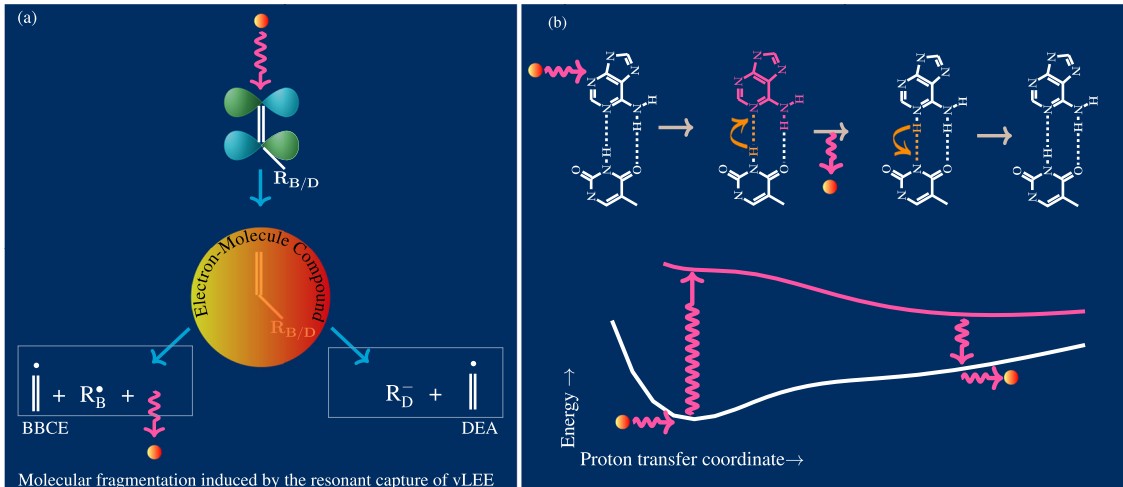

**Fig. 3 The vLEE-induced molecular fragmentation reactions and the stability mechanism available in base pairs against such molecular damages. (a)** It shows molecular fragmentation reactions induced due to the resonance capture of a vLEE into its $\pi^*$ orbital. Depending on the functional group R, the resonantly formed electron-molecule compound undergoes a neutral fragmentation (BBCE) or negative-ion fragmentation (DEA) reaction. **(b)** This shows schematic representation of the protective mechanism available to the complementary base pairs against the resonant capture of a vLEE into a $\pi^*$ orbitals of its base moiety. The vLEE-captured base moiety and the potential energy of the resulting negative ion resonance state of the base pair are shown in pink, while the energy curve of the corresponding neutral ground state is shown in white. Once the vLEE is resonantly attached to one of the base moieties, an ultrafast proton transfer from its complementary base helps to eject the resonantly captured vLEE from the molecular field of the base pair, thus rapidly recovering the neutral ground state before undergoing any molecular fragmentation reaction.

approximately two orders of magnitude smaller in double-helix DNA[44] than in oligonucleotides[45]. This leads naturally to the conclusion that a rapid biomolecular mechanism based on base pairing protects the structural integrity of the DNA from damage induced by vLEEs. The author has recently demonstrated that molecular damage caused to a base pair by the $\pi^*$-resonant capture of vLEE is avoided by the transfer of a proton between pairs through a hydrogen bond[46]. When a vLEE is captured resonantly in the $\pi^*$ orbital of a purine base moiety, i.e., guanine or adenine, its complementary base transiently and rapidly transfers a hydrogen bonded proton to the electron-attached nucleobase. Such rapid interpair transfer of a hydrogen bonded proton, which is one of the fastest non-damaging nuclear relaxation processes know, not only delocalizes excess negative charge throughout the molecular domain of the base pair, but also dissipates the electronic energy of the vLEE into its nuclear modes and, therefore, electronically metastabilizes the corresponding NIRS. This electronic metastabilization eliminates structural damage caused by $\Pi^* - \Sigma^*$ state-crossing. The harmful excess electron then gets autodetached from the corresponding metastable minimum of the NIRS (see Fig. 3b). Upon autodetachment of the excess electron, the equilibrium ground state of the neutral base pair is seamlessly restored by the reverse transfer of the proton. Thus, this remarkable interpair proton transfer function of complementary base pairing, which also provides excellent stability to DNA against $\pi - \pi^*$ photoexcitation-induced damages, renders the scaffold to prevent direct damage caused by $\pi^*$ type vLEE capture in purine bases. On the other hand, when a vLEE is captured in a pyrimidine moietiy, a molecular relaxation in the Frank-condon region of the base pair itself creates a metastable minimum and, hence, the captured electron is autodetached without transferring an interpair proton[46].

**Chemical evolution and improvement of UIS mechanisms**. An enhanced UIS of the complementarily paired bases to survive UV and vLEE intense adverse physicochemical conditions rather than the individual bases may have enabled the proliferation of nucleobases as complementary pairs under extremely unfavorable prebiotic conditions. In view of the existence of an intense prebiotic UV condition, this enhanced UIS of complementary base pairs can be attributed to their proliferation and their chemical evolution as basic units of the genetic code. The structural and chemical composition of the advanced molecules must have evolved in accordance with the enhancement of their UIS mechanisms. Accordingly, we discuss here the most likely molecular mechanisms of UIS involving the chemically advanced forms of base pairs.

**Multiple assaults and enhanced UIS of complementary nucleotides**. Complementary pairing provides intrinsic stability to nucleobases against UV photons and vLEEs. But that stability can only safeguard the pair from local attacks on one nucleobase at a time. Under intense shortwave UV conditions, a highly destructive situation can occur naturally when the two nucleobases of the complementary nucleotides are photoexcited or both capture vLEEs. Since the complementarity that begins with the base pair has evolved into DNA, an examination of the broad DNA segment and hydrated chemical medium is necessary to understand their proliferation against such multiple assaults. Here, we consider two situations where the two nucleobases of a complementary nucleotide segment of DNA–comprising sugar-phosphate units on either side of the complementary base pair–are either simultaneously photoexcited or undergo resonant electron attachment. When such photoexcited or electron captured states are created in hydrogen-bonded molecular moieties,

they collectively and concertedly relax by ionizing an electron through a ultrafast electronic relaxation mechanism known as intermolecular Coulomb decay (ICD)[10–16,32]. Due to the wave-function overlap between the nucleobases in the complementary pair, the ICD becomes the most effective relaxation channel to dispose the photo energy/captured electrons. The neutral electronic ground state of the complementary nucleotide is then restored with the aid of a hydrated chemical environment and a sugar phosphate backbone. One may note that an ionized nucleobase can, in principle, be neutralized by proton-coupled electron transfer from the medium[47] or from the adjacent nucleobase that is not complementary to the ionized nucleobase[48]. Subsequent stabilization of the transferred proton may prevent its reversal to its original position, leading to DNA damage. However, in complementary paired bases, the hole can be internally and competitively localized within the pair itself by the exchange of protons of the interpair hydrogen bonds. Furthermore, the electron-rich DNA backbone, its conductivity, and the availability of the hydrated electron make charge neutralization very competitive and fast. Schematic illustrations of the unified intrinsic stability based on ICD mechanism against molecular damage caused by photoexcited multiple moieties and electron-captured multiple moieties are shown in Figs. 4 and 5, respectively.

**Role of the molecular environment in advanced UIS mechanisms**. The UIS mechanisms discussed here are non-local relaxation channels opened by the proximity of a neighboring molecular unit, and the complementary base pairing ensures mutual neighboring units for nucleobases. The UIS mechanism that ensures photostability and stability against vLEEs is a molecular process involving interpair proton transfer. Alternatively, a hydrogen-donating intermolecular relaxation channel can be opened by the cellular hydrated medium. Experimental evidence that microhydration suppresses vLEE-induced fragmentation of certain nucleobases[49] and the suppression of electron-induced strand break under wet conditions of DNA[50] are relevant here. Although the role of proton transfer has not been attributed to the observed phenomenon, a remarkable inter-moiety proton transfer-assisted suppression of single strand breaks has recently been reported in the literature[50]. The experimental and theoretical demonstration of ICD phenomenon in water[13,51], hydrogen bonded systems[15,16] and $\pi$-stacked molecules[14] clearly suggests that the UIS mechanisms based on ICD phenomena against multiple assaults can also operate beyond the base pairs. The water in the cellular medium and the $\pi$-stacked near-neighbors in DNA could be playing fundamental yet hitherto unknown role in UIS mechanisms when the structural integrity of DNA is affected by shortwave photons and vLEEs. In this way, the investigation of hitherto unknown stability inherent in the basic building units and their chemically evolved biomolecules can reveal the precise and subtle physicochemical conditions of a prebiotic medium in which they have chemically evolved.

**Extent of UIS mechanisms**. Pairing by complementarity is the simplest and most basic structural rearrangement that protects nucleobases by opening a new UIS channel against the co-existing shortwave UV photons and vLEEs. In fact, such restructuring of prebiotic molecules, which had to evolve chemically and enables the molecules to dissipate the absorbed UV energy, is argued to be the vital step in the chemical evolution of advanced molecules[52–54]. The UIS mechanism, which we adduce here as the origin of biomolecular complementarity, provides a general and more inclusive reason, not limited to the direct effects of UV

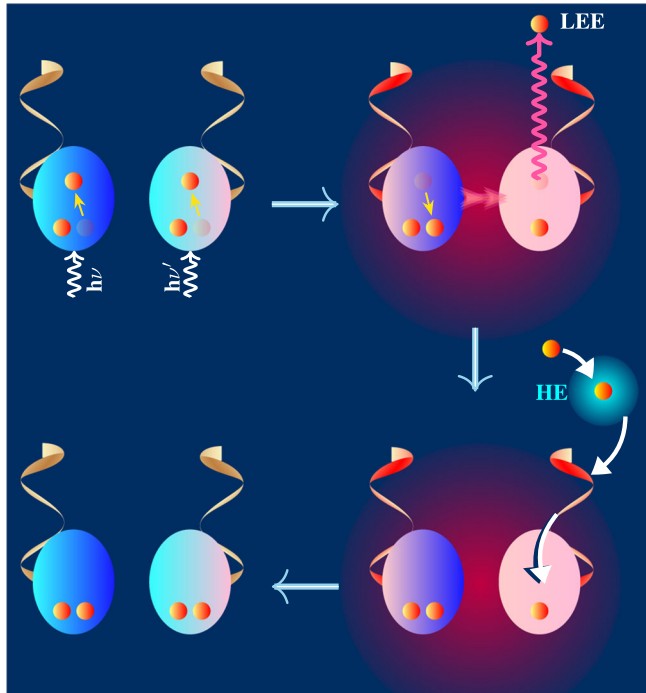

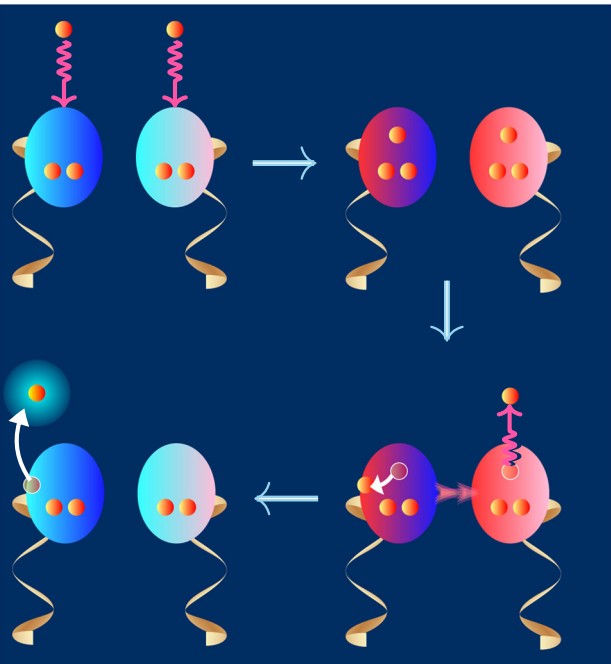

**Fig. 4 Non-radiative energy decay following simultaneous multiple photo-excitation of complementary nucleotides in a medium.** The oval structure and the Möbius ribbon represent the nucleobase and the sugar-phosphate components of the nucleotide moiety in DNA, respectively. The small orange circles inside the nucleobase represent the $\pi$ electrons under consideration. Simultaneous photoexcitation of both bases of complementary nucleotides results in simultaneous relaxation of both nucleobases by emission of a LEE through the ICD process. The phosphate group, owing to the water solubility, forms a hydrated shell around them[55], and this opens the door for the sugar-phosphate group to recover the hydrated electron (HE), i.e., the hydrated LEE. Finally, the hole created in the nucleobase due to ICD is annihilated by electron transfer from the negatively charged sugar-phosphate backbone of DNA. The intra- and inter-molecular vibrational energy transfer and solvation of the LEE dissipate the absorbed photon energy. The resulting hot nucleotides and the medium are represented by a fading red color.

**Fig. 5 A molecular mechanism that removes two electrons attached to the bases of a complementary nucleotide.** After LEEs are captured onto both bases of complementary nucleotides, one of them is rapidly removed by the ICD process. In this process, the second electron is de-excited into a dipole-bound orbital. Such electronic de-excitation from the negative ion resonance state to the dipole-bound anionic state has been reported[56,57]. This loosely bound anionic electron then dissociates into the continuum of the cellular medium via an ultrafast rotation-to-electronic or vibration-to-electronic nonadiabatic energy transfer[58–60].

photons, for the dissipative restructuring. The existence of UIS mechanism in biomolecules suggest that among the prebiotic molecules that were synthesized, only those that consolidate internal stability against all the coexisting molecular deleterious agents that exacerbate harsh prebiotic conditions have proliferated and thus progressed in chemical evolution.

**Biological consequences of UIS mechanisms**. What emerges clearly from the observations thus far is that the development and refinement of UIS mechanisms in a harsh prebiotic medium fueled by the absence of a stratospheric ozone layer proliferated nucleobases into canonical complementary pairs. Further, the chemical evolution of polymeric nucleic acids must have begun with the increasing prevalence of canonical complementary pairs over time on the prebiotic Earth. The most important biological consequence of the nucleobases proliferating as complementary pairs due to their enhanced UIS in a prebiotic medium is that it seeds molecular-level complexity necessary for biological functions such as recognition, regulation and replication. The understanding that the prebiotic world, which was not protected by the stratospheric ozone layer, was driven by chemical evolution based on complementary pairs supports the conventional

conceptualization of the origin of life based on DNA dominance. In other words, if the ozone layer had existed in prebiotic times, life would likely have evolved very differently from what we see today.

**Summary and outlook**. In the absence of a protective stratospheric ozone layer, a prebiotic ambient physicochemical medium intensified by the coexistence of shortwave UV photons and vLEEs must have existed on early Earth. The chemical evolution of advanced biological molecules must have started with the synthesis of their building units, i.e., nucleobases, by the energy sources of this prebiotic medium and continued through their increased proliferation over time. The persistence of the combined intrinsic stability of nucleobases in the prebiotic environment against the continuous assaults by the ubiquitous photons and the vLEEs may have given them an advantage for their proliferation. Pairing by complementarity is the simplest yet most efficient step in the chemical evolution of the nucleobases that enhance their UIS against the harsh conditions of this prebiotic medium. The enhanced UIS achieved due to complementary pairing may have enabled the proliferation of nucleobases as complementary pairs under these extremely unfavorable prebiotic conditions. The observation of ultrafast deactivation of the photoexcited states and autoionization of the negative ion resonance states specific to the base pairs competing fast enough with the corresponding relaxation pathway of the isolated nucleobases confirms that the complementary base pairs advance the chemical evolution. It can, therefore, be concluded that complementary base pairing is a manifestation of a harsh prebiotic medium in which nucleobases have chemically evolved. The subtle expression of that harsh prebiotic medium fueled by the absence of a

protective stratospheric ozone layer is inherent in the chemically advanced molecules of life, and the available evidence exhorts man to seek them.

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

## Acknowledgements
The author is grateful to Prof. Léon Sanche and Prof. E. Krishnakumar for discussions regarding electron–biomolecular interactions and insights into biomolecular stability mechanisms. The author is greatly indebted to Prof. Lorenz Cederabaum and Prof. Aravind Gopalan for stimulating discussions on the intermolecular Coulombic decay processes.

## Competing interests
The author declares no competing interests.
