## [Peer Review File · Communications Chemistry]

Reviewers' comments:

Reviewer #1 (Remarks to the Author):

This manuscript studied theoretically the stability mechanisms of nucleic acid base pairs in an environment where the UV light or low-energy electrons irradiations exist. In comparison to individual nucleobase, nucleobase pairs can dissipate the UV photon energy through a fast intermolecular Coulombic decay (ICD) process, thus preventing the dissociation of biomolecules. The photostability mechanism proposed by the author may be one of the reasons why bases evolved in the form of complementary base pairs. The double-photon excitation and double-electron attachment-induced ICD mechanisms are interesting, however, there are lacks of quantitative descriptions about the real or absolute contributions of these processes, which need to be clarified.

More questions and comments are:

1. In the discussion about the ICD process induced by double-photon-excited of nucleobase pairs, the ionized nucleobase can restore the neutral state by transferring an electron from the negatively charged sugar-phosphate backbone backbone of DNA. In addition to the electron transfer process, the ionized nucleobase can rapidly transfer a proton to neighboring base (Nat. Chem. 4, 323(2012); J. Am. Chem. Soc. 135, 3904(2013)), which is an underlying competing mechanism with the electron transfer process leading to DNA mutations. Discussions on this topic are needed.
2. in the section of "Stability against vLEEs", it is proposed in ref [48] that the excess energy generated by molecular vibration de-excitation leads to the ionization of neighboring anions, contrary to the electron de-excitation process stated by the author. Is there any other evidence supporting that weakly bound electrons can de-excite to more stable dipole-bound orbitals? The author needs to clarify this.
3. In previous photoelectron experiments of the uracil-alanine anionic complex (J. Chem. Phys. 120, 6064 (2004)), it has been demonstrated that electron attachment can induce intermolecular proton transfer, leading to the formation of stable molecular complexes. In the mechanism illustrated in Figure 3, what is the driving force behind the hydrogen atom returning to its original position?
4. The conically intersecting between ground state, $\pi-\pi^*$ state and $\pi-\sigma^*$ state prevent the nucleobase from dissociation. The presence of water as solvent molecules removes the conical intersection of $\pi-\sigma^*$ with the ground state. Instead of IC to the electronic ground state, an excited-state hydrogen-transfer reaction takes place in nucleobase with water [26]. In a real biological environment, the presence of aqueous solutions may catalyze mutations in nucleic acid bases. It is suggested to add relevant discussions on this topic.
5. For the UV light experiments in refs. [10, 31], the emitted electrons and ions are measured separately. This means that the measured LEEs are not correlated with the ions. The UV light can generate many LEEs from metal surface of the chamber and the spectrometer, etc. Therefore, some coincident experiments are necessary to demonstrate those ICD mechanisms. The author should point out this in the manuscript.

6. The central focus of the author's theory lies in the double-photon excitation and double-electron attachment-induced ICD reactions. While the proposed theory is indeed fascinating, there is a scarcity of relevant theoretical calculations and experimental evidence. To make the argument more convincing to readers, it would be preferable to incorporate additional experiments and more in-depth theoretical calculations.

Reviewer #2 (Remarks to the Author):

In this Perspective, the author tries to justify his opinion according to which the complementary hydrogen bonds between nucleobases is one of the main protective factors that allowed the complex life molecules to survive under very harsh conditions that prevailed on prebiotic Earth. Namely, without the ozone layer, those molecules were exposed to a significant intensity of shortwave UV photons. Moreover, the UV photons absorbed by interacting π -molecular systems led to the release of very low-energy electrons (vLEEs). Thus, UV photons and vLEEs were the main damaging species on prebiotic Earth.

The remaining part of the article is devoted to the relation between the complementary hydrogen bonds that are formed by nucleobases and their increased resistance toward UV photons and vLEEs. The article is written clearly and logically. Its novelty is related to the perspective in which complementary hydrogen bonds are presented. Namely, they are not shown as a genetic code source but rather as an origin of the exceptional resistance of nucleobases to damaging species.

I have only several minor remarks that should be addressed before this article can be published:

- The author writes: "Deposition in excess of 3eV energy on a base in DNA due to absorption of UV radiation, which populates the $\pi \rightarrow \pi^*$ excited state of nucleobases, in principle, can trigger a variety of photochemical reactions in the DNA leading to mutagenesis and carcinogenesis". However, the absorption of DNA bases at 3 eV is negligible – see e.g. *Phys. Chem. Chem. Phys.*, 2010,12, 4959-4967.
- Please add a reference to the listed lifetimes "...the excited state lifetime ultra-short (A:1.0ps; G: 0.8ps, T: 6.4ps; C: 3.2ps).
- The author suggests that purine-purine or pyrimidine-pyrimidine pairs are not formed due to the shape of proton transfer potential: "On the other hand, in non-canonical symmetric pairs, i.e., pyridine-pyridine pairs and purine-purine pairs, the corresponding proton transfer path is symmetric double well and does not cross with the neutral ground state." Although it may be one of the reasons, this also seems to be an exaggeration. Purine-pyrimidine complementarity is justified by the geometry of the double-stranded helix and it seems to be a primary reason for which Nature chose such a pattern.
- "However, the low energy N-glycosidic σ^* anti-bonding orbital of the nucleotide can cross with the π^* orbital for a short stretch of the corresponding bond. As a result, the resonantly captured π^* electron can easily get transferred to the σ^* orbital.", The electron captured to nucleobase is preferably transferred to the σ^* anti-bonding orbital of the CX'-O bond (where X=5 or 3) rather than to the σ^* N-glycosidic bond (*Radiation Research*, 2006, 165(6), pp. 721–729).
- "The UIS mechanism that ensures photostability and stability against vLEEs is a molecular process involving interpair proton transfer." It is well known that electron attachment to the GC base pair induces proton transfer (*Journal of the American Chemical Society*, 2009, 131(7), pp. 2663–2669). A similar process does not occur in the AT base pair. This fact should be briefly discussed in the reviewed

article.

Minor language issues:

- "...The fact that copious amounts amounts of vLEEs are naturally produced by UV..." – remove the redundant "amounts".
- "...This implied that the preventive mechanisms against the structural caused by shortwave UV photons..." – "the structural" what?
- Correct English, please: "...ICD phenomena against multiple assaults can also operative beyond the base pairs..."

Author response to the comments of Reviewer #1

1. *In the discussion about the ICD process induced by double-photon-excited of nucleobase pairs, the ionized nucleobase can restore the neutral state by transferring an electron from the negatively charged sugar-phosphate backbone backbone of DNA. In addition to the electron transfer process, the ionized nucleobase can rapidly transfer a proton to neighboring base (Nat. Chem. 4, 323(2012); J. Am. Chem. Soc. 135, 3904(2013)), which is an underlying competing mechanism with the electron transfer process leading to DNA mutations. Discussions on this topic are needed.*

We have included the following discussion on this possibility in the revised manuscript.

One may note that an ionized nucleobase can, in principle, be neutralized by proton-coupled electron transfer (PCET) from the medium [48] or from the adjacent nucleobase that is not complementary to the ionized nucleobase [49]. Subsequent stabilization of the transferred proton may prevent its reversal to its original position, leading to DNA damage. However, in complementary paired bases, the hole can be internally and competitively localized within the pair itself by the exchange of protons of the interpair hydrogen bonds. Furthermore, the electron-rich DNA backbone, its conductivity, and the availability of the hydrated electron make charge neutralization very competitive and fast. (Page 17)

2. *In the section of “Stability against vLEEs”, it is proposed in ref [48] that the excess energy generated by molecular vibration de-excitation leads to the ionization of neighboring anions, contrary to the electron de-excitation process stated by the author. Is there any other evidence supporting that weakly bound electrons can de-excite to more stable dipole-bound orbitals? The author needs to clarify this.*

We have added the following two references [Ref 51 and 52 in the revised manuscript] where a negative-ion-resonance-state to dipole-bound-anionic-state transition is reported in connection with radiative electron attachment (REA).

- 1) F. Carelli; F. A. Gianturco; R. Wester; M. Satta J. Chem. Phys. **141**, 054302 (2014) <https://doi.org/10.1063/1.4891300>
 - 2) Wojciech Skomorowski , Sahil Gulania and Anna I. Krylov Phys. Chem. Chem. Phys. **20**, 4805-4817 (2018) DOI: 10.1039/C7CP08227D
3. *In previous photoelectron experiments of the uracil-alanine anionic complex (J. Chem. Phys. 120, 6064 (2004)), it has been demonstrated that electron attachment can induce intermolecular proton transfer, leading to the formation of*

stable molecular complexes. In the mechanism illustrated in Figure 3, what is the driving force behind the hydrogen atom returning to its original position?

Unlike nucleobases, the alanine molecule supports stable covalent anionic states. Therefore, when an electron is captured to the nucleobase moiety of the nucleobase-alanine pair, a stable anionic state is formed by the interpair proton transfer from the alanine moiety to the nucleobase moiety. Although such proton transfer in nucleic acid base pairs stabilizes the corresponding negative ion resonance state, it does not result in the formation of a thermodynamically stable covalent anion. Therefore, the captured electron gets autodetached from the stabilized negative ion resonance state of the base-pair and the resulting proton-transferred nucleobase pair is relaxed barrierlessly to its thermodynamically stable neutral ground state.

4. *The conically intersecting between ground state, $\pi - \pi^*$ state and $\pi - \sigma^*$ state prevent the nucleobase from dissociation. The presence of water as solvent molecules removes the conical intersection of $\pi - \sigma^*$ with the ground state. Instead of IC to the electronic ground state, an excited-state hydrogen-transfer reaction takes place in nucleobase with water [26]. In a real biological environment, the presence of aqueous solutions may catalyze mutations in nucleic acid bases. It is suggested to add relevant discussions on this topic.*

We have added the following discussion in the revised manuscript.

The protective function of the $^1\pi\pi^*$ specific to the base pair becomes more relevant in a protic medium. Although isolated nucleic acid bases are intrinsically photostable, the presence of an aqueous environment may prevent the ground state recovery of its π^* excited state, where the $^1\pi\sigma^*$ states promote a hydrogen-transfer process from the photoexcited nucleobase to the protic solvent [26]. Subsequent burial of the transferred hydrogen in the solvated medium can lead to molecular damage. However, when the chromophore base is complemented in a base pair through intermolecular hydrogen bonds, the ground state recovery via a newly created $^1\pi^*\pi^*$ path along the interpair hydrogen transfer can be competitive with the hydrogen atom's burial into the medium. This may be another reason why base pairs evolved as the basic units of the genetic code. (Page 10)

The author thanks the reviewer for this comment.

5. *For the UV light experiments in refs. [10, 31], the emitted electrons and ions are measured separately. This means that the measured LEEs are not correlated with the ions. The UV light can generate many LEEs from metal surface of the chamber and the spectrometer, etc. Therefore, some coincident experiments are necessary to demonstrate those ICD mechanisms. The author should point out this in the manuscript.*

We revised the manuscript as

In the reported ICD experiments of Ref. 10 and Ref. 32, LEEs and their counterpart cations produced by π -molecules under ambient UV light via the ICD mechanism were confirmed by a set of experiments that rule out the role of photogenerated electrons, rather than a coincidence measurement of the electron-cation pair. (Page 12)

- 6. The central focus of the author's theory lies in the double-photon excitation and double-electron attachment-induced ICD reactions. While the proposed theory is indeed fascinating, there is a scarcity of relevant theoretical calculations and experimental evidence. To make the argument more convincing to readers, it would be preferable to incorporate additional experiments and more in-depth theoretical calculations.*

The author also fully agrees with *Reviewer* that in-depth theoretical and experimental studies are desirable to advance the proposed theory. As a first step in this direction, the author himself has focused his current work on the two-electron attachment in base pairs. The author is very optimistic that this article and his continued research work towards this direction will encourage chemists and physicists studying biologically relevant molecules to view their research data from a prebiotic perspective as well.

Author response to the comments of Reviewer #2

1. *The author writes: “Deposition in excess of 3eV energy on a base inDNA due to absorption of UV radiation, which populates the $\pi - \pi^*$ excited state of nucleobases, in principle, can trigger a variety of photochemical reactions in the DNA leading to mutagenesis and carcinogenesis”. However, the absorption of DNA bases at 3 eV is negligible - see e.g. Phys. Chem. Chem. Phys., 2010,12, 4959-4967.*

We have revised this sentence as follows

Absorption of UV radiation below 320nm wavelength by a DNA nucleobase, which populates its $\pi \rightarrow \pi^*$ excited state, can trigger a variety of photochemical reactions in the DNA leading to mutagenesis and carcinogenesis. [Page 7]

2. *- Please add a reference to the listed lifetimes “...the excited state lifetime ultra-short (A:1.0ps; G: 0.8ps, T: 6.4ps: C: 3.2ps).*

We have added the reference (Ref No 25) for the lifetime data. (Page 9)

3. *The author suggests that purine-purine or pyrimidine-pyrimidine pairs are not formed due to the shape of proton transfer potential: “On the other hand, in non-canonical symmetric pairs, i.e., pyridine-pyridine pairs and purine-purine pairs, the corresponding proton transfer path is symmetric double well and does not cross with the neutral ground state.” Although it may be one of the reasons, this also seems to be an exaggeration. Purine-pyrimidine complementarity is justified by the geometry of the double-stranded helix and it seems to be a primary reason for which Nature chose such a pattern.*

The revised manuscript now sets out the context of the discussion as follows

Although purine-pyrimidine complementarity is justified by the geometry of the double-stranded helix, in light of the reports that duplex structures suitable for information storage and replication fundamental to molecular evolution are possible from purine bases alone [31], the question of why Nature does not select such complementary structures becomes relevant. (Page 11)

4. *“However, the low energy N-glycosidic σ^* anti-bonding orbital of the nucleotide can crosses with the π^* orbital for a short stretch of the corresponding bond. As a result, the resonantly captured π^* electron can easily get transferred to the σ^* orbital.”., The electron captured to nucleobase is preferably transferred to the σ^* anti-bonding orbital of the CX'-O bond (where X=5 or 3) rather than to the σ^* N-glycosidic bond (Radiation Research, 2006, 165(6), pp. 721-729).*

We revised the text as follows

As a result, the resonantly captured π^* electron can easily get transferred to the σ^* orbital or competitively to the σ^* orbital of the CX'-O bond (where X=5 or 3)[40]. (Page 13)

5. *“The UIS mechanism that ensures photostability and stability against vLEEs is a molecular process involving interpair proton transfer.” It is well known that electron attachment to the GC base pair induces proton transfer (Journal of the American Chemical Society, 2009, 131(7), pp. 2663–2669). A similar process does not occur in the AT base pair. This fact should be briefly discussed in the reviewed article.*

In the case of resonant capture of vLEE by a purine base, i.e., guanine or adenine, its complementary base rapidly transfer an interpair proton and metastabilizes the resulting electron attached purine moiety. The harmful excess electron is autodetached from this resulting metastable minimum of the proton transferred basepair. On the other hand, when a vLEE is captured on the pyrimidine moiety, a molecular relaxation in the Frank-condon region of the base pair itself creates a metastable minimum and, hence, the captured electron is autodetached without any proton transfer. This fact is now evident in the revised manuscript. (Page 15)

6. *Minor language issues:*
 - *“... The fact that copious amounts amounts of vLEEs are naturally produced by UV...” – remove the redundant “amounts”.*
 - *“... This implied that the preventive mechanisms against the structural caused by shortwave UV photons...” – “the structural” what?*
 - *Correct English, please: “...ICD phenomena against multiple assaults can also operative beyond the base pairs...”*

These corrections were made in the revised manuscript

REVIEWERS' COMMENTS:

Reviewer #1 (Remarks to the Author):

The authors have carefully and convincingly addressed most comments and concerns raised by the referees. The quality of the revised manuscript has been improved, and I also confirm that this is a nice and well-written paper. I am happy to recommend publication of the current version of the manuscript in Communications Chemistry.

Reviewer #2 (Remarks to the Author):

The author addressed the reviewer's remarks convincingly. Now, the paper can be published as is.